# Identification of a Potential High-Risk Clone and Novel Sequence Type of Carbapenem-Resistant *Pseudomonas aeruginosa* in Metro Manila, Philippines

**DOI:** 10.3390/antibiotics14040362

**Published:** 2025-04-01

**Authors:** Sherill D. Tesalona, Miguel Francisco B. Abulencia, Maria Ruth B. Pineda-Cortel, Sylvia A. Sapula, Henrietta Venter, Evelina N. Lagamayo

**Affiliations:** 1The Graduate School, University of Santo Tomas, Espana Blvd., Manila 1015, Philippines; 2Department of Medical Technology, Faculty of Pharmacy, University of Santo Tomas, Espana Blvd., Manila 1015, Philippines; mbpineda-cortel@ust.edu.ph; 3Advanced Molecular Technologies Laboratory, Research Institute for Tropical Medicine, Muntinlupa City 1781, Philippines; abulencia.miguel@gmail.com; 4Research Center for the Natural and Applied Sciences, University of Santo Tomas, Espana Blvd., Manila 1015, Philippines; 5Health and Biomedical Innovation, Clinical Health Sciences, University of South Australia, Adelaide 5000, Australia; sylvia.sapula@unisa.edu.au (S.A.S.); rietie.venter@unisa.edu.au (H.V.); 6Institute of Pathology, St. Luke’s Medical Center, Quezon City 1112, Philippines; enlagamayo@gmail.com; 7Department of Clinical Pathology, University of Santo Tomas Hospital, Manila 1015, Philippines; 8Institute of Pathology, Chinese General Hospital and Medical Center, Manila 1014, Philippines

**Keywords:** antimicrobial resistance, whole-genome sequence, sequence type, carbapenemase, KPC-2, VIM-2, OXA-74

## Abstract

Carbapenem-resistant *Pseudomonas aeruginosa* (CRPA) is a significant opportunistic human pathogen, posing a considerable threat to public health due to its antimicrobial resistance and limited treatment options. The incidence of CRPA is high in the Philippines; however, genomic analysis of CRPA in this setting is limited. Here, we provide the phenotypic and molecular characterization of 35 non-duplicate CRPA obtained from three tertiary hospitals in Metro Manila, Philippines, from August 2022 to January 2023. Six sequence types (STs), including international high-risk clones ST111 and ST357, were identified. This article highlights the first report in the Philippines on the identification of *P. aeruginosa* harboring *Klebsiella pneumoniae* Carbapenemase-2 (KPC-2), coproduced with Verona Integron-encoded Metallo-beta-lactamase-2 (VIM-2) and Oxacillinase-74 (OXA-74). Notably, this is also the first report of KPC in the Philippines identified in *P. aeruginosa*. New Delhi Metallo-beta-lactamase-7 (NDM-7), coproduced with Cefotaxime-Munich-15 (CTX-M-15) and Temoneira-2 (TEM-2), was also identified from a novel ST4b1c. The relentless identification of NDM in the Philippines’ healthcare setting poses a significant global public health risk. The initial detection of the *P. aeruginosa* strain harboring KPC exacerbated the situation, indicating the inception of potential dissemination of these resistance determinants within *P. aeruginosa* in the Philippines.

## 1. Introduction

Carbapenem-resistant *P. aeruginosa* (CRPA) is classified by the World Health Organization (WHO) as a “high priority” pathogen in its 2024 Bacterial Priority Pathogens List [1]. *P. aeruginosa* is an opportunistic pathogen [2], notorious for causing severe infections, particularly in healthcare settings and among individuals with compromised immune systems [3,4]. The rapid dissemination of CRPA strains not only jeopardizes patient outcomes but also poses a significant challenge to global healthcare systems. The main mechanisms by which *P. aeruginosa* can develop resistance to carbapenems include the production of enzymes that can degrade carbapenem antibiotics, such as carbapenemases [5]; the overexpression of drug efflux pumps, such as those in the resistance-nodulation-division (RND) family, which help the cell export carbapenems [4]; and decreased outer membrane permeability through the downregulation or loss of outer membrane porin D (OprD), which are required for carbapenem entry [6,7,8]. Carbapenemases are enzymes that can hydrolyze carbapenem antibiotics, rendering them ineffective [9,10]. Carbapenemases are classified into two families: serine and metallo-beta-lactamases (MBLs) [11]. Serine beta-lactamases such as class A and class D beta-lactamases require serine residue at the active site [12]. KPC-2 is a class A beta-lactamase typically found in Enterobacteriaceae such as *Klebsiella pneumoniae*, hydrolyzing a broad range of beta-lactams, including carbapenems, and contributing to global multidrug resistance [1]. MBLs, such as New Delhi metallo-beta-lactamases (NDM) and Verona-Integron metallo-beta-lactamases (VIM), are amber class B enzymes that require one or two zinc ions for catalytic activity [12]. Unlike serine beta-lactamases, MBLs are not inhibited by typical therapeutic inhibitors such as clavulanic acid, tazobactam, or sulbactam [13]. VIM-2 is a class B metallo-beta-lactamase, often seen in *P. aeruginosa*, capable of hydrolyzing carbapenems and frequently co-occurring with other resistance genes [2]. Finally, OXA-74 is a class D beta-lactamase associated with carbapenem resistance, particularly in *Acinetobacter* species, spreading via mobile genetic elements [3,4,5]. The acquisition of carbapenemases is critical because these enzymes can hydrolyze most beta-lactam antibiotics [14]. Carbapenemase genes are often carried on mobile genetic elements, such as plasmids, transposons, and insertion sequences, which facilitate their transfer to other bacterial species [10,15].

The incidence of CRPA in the Asia-Pacific region has been reported, and the four countries with the highest incidences were Indonesia (50.45%, 112/222), India (23.08%, 9/39), Italy (17.72%, 45/254), and China (14.81%, 8/54) [16,17]. Several countries in Southeast Asia, including Bangladesh, India, Indonesia, Nepal, Sri Lanka, and Thailand, have reported a worrying trend of an increase in AMR, while the Philippines was not mentioned in that report [18]. However, the AMR Surveillance Reference Laboratory annual report data summary indicates that, in 2022, there were 7,561 recorded *P. aeruginosa* isolates, with 15.4% and 12.8% exhibiting resistance to the carbapenem antibiotics, imipenem, and meropenem respectively [19]. Rates are rising in 2023, wherein there was a reported increase in the total number of *P. aeruginosa* isolates, reaching 9,680. This was accompanied by a rise in imipenem resistance to 18.97% and meropenem resistance to 15.12% [20]. Despite the high incidence of CRPA and the critical need for new antimicrobial agents against this pathogen, there is a scarcity of research regarding the molecular attributes of CRPA strains in the Philippines setting.

Continued research is paramount to safeguarding public health against the growing threat posed by CRPA by providing current information. Given that antimicrobial resistance (AMR) and CRPA are global concerns that significantly impact our health and economy, a better understanding of the molecular characteristics of this critical pathogen is needed to reduce the risk of dissemination. Hence, this study reports on the surveillance of CRPA from three tertiary hospitals in the Philippines using antibiotic susceptibility test (AST) and whole-genome sequencing (WGS) to identify resistance genes and understand the resistance mechanisms.

## 2. Results

### 2.1. Information on Sources of Isolates

The 35 non-duplicate CRPA isolates investigated were unevenly distributed across the three hospitals: 8 from Hospital A, 16 from Hospital B, and 11 from Hospital C, corresponding to 22.9%, 45.7%, and 31.4%, respectively. The distribution of eight non-duplicate CRPA isolates from Hospital A, categorized by hospital wards and specimen sources, is as follows: 62.5% (n = 5/8) were collected from medical wards, and 12.5% (n = 1/8) were from the pediatric ward, surgery, and critical care unit (CCU). Additionally, 62.5% (n = 5/8) were obtained from respiratory specimens (endotracheal aspirate, sputum, and bronchoalveolar lavage), 25.0% (n = 2/8) from blood, and 12.5% (n = 1/8) from bone marrow aspirate. Hospital B reported a total of 16 non-duplicate CRPA isolates, with 25.0% (n = 4/16) obtained from medical wards and 75.0% (n = 12/16) from intensive care units (ICU). Among these, 56.25% (n = 9/16) were sourced from respiratory specimens, 12.5% (n = 2/16) from gastric aspirate, and 6.25% (n = 1/16) each from wound tissue, catheter drain, and ascitic fluid. Hospital C provided 36.4% (n = 4/11) non-duplicate CRPA isolates sourced from medical wards, 27.3% (n = 3/11) from obstetrics and gynecology, another 27.3% (n = 3/11) isolates from ICU, and 9.1% (n = 1/11) from surgery. These isolates were derived from 90.9% (n = 10/11) respiratory specimens and 9.1% (n = 1/11) from gastric aspirates. Finally, the 35 non-duplicate CRPA isolates were obtained from a cohort of patients aged 14 to 93 years, 57.0% of which were male (Appendix A).

### 2.2. Antimicrobial Susceptibility of the CRPA Isolates

Analysis revealed that 62.5% (n =10/16) CRPA isolates obtained from Hospital B were resistant to ceftazidime, cefepime, and ciprofloxacin, while 72.7% (n = 8/11) CRPA isolates obtained from Hospital C were resistant to both ceftazidime and cefepime, and 90.9% (n = 10/11) were resistant to ciprofloxacin, as shown in Figure 1A. Moreover, a low proportion of CRPA isolates obtained from Hospital A were resistant to ceftazidime, cefepime, and ciprofloxacin. Only 37.5% (n = 3/8) of the CRPA isolates from Hospital A were resistant to ceftazidime, and 25.0% (n = 2/8) were resistant to cefepime and ciprofloxacin. The incidence rate and resistance level against ceftazidime-avibactam and colistin cannot be compared among the three hospitals because the institutions have different protocols for testing multidrug-resistant organisms (MDROs) against these antibiotics. Of the three hospitals, only Hospital C automatically tested CZA and COL against MDROs and reported them following the cascade reporting rules and upon physician request. Notably, each strain of *P. aeruginosa* obtained from different wards of each hospital revealed diverse results of antibiogram and STs as shown in Appendix A. The 35 CRPA isolates obtained from Hospital A, B and C are all (100.0%) resistant to imipenem, However, of the 35 CRPA isolates, 14.3% (n = 5/35) showed intermediate AST results against meropenem (Figure 1A). Additionally, the 35 isolates investigated showed resistance to at least one antibiotic in three or more antimicrobial classes tested as shown in Figure 1B. Moreover, the CRPA isolates from hospitals B and C showed resistance to a greater number of different antibiotics. The MDR phenotype was observed in 100.0% (n = 11/11) of the CRPA isolates obtained from Hospital C, 75.0% (n = 12/16) from Hospital B, and 87.5% (n = 7/8) from Hospital A (Figure 1C and Appendix A). Varying resistance profiles of each hospital are due to factors such as local antibiotic use from each hospital, patient demographics, and infection control practices of the three hospitals.

Following the CLSI breakpoints, the phenotypic analysis revealed the following incidence of resistance across the three hospitals: IMP, 100.0% (n = 35/35); MEM, 82.9% (n = 29/35); CAZ, 60.0% (n = 21/35); FEP, 57.1% (n = 20/35); TZP, 48.6% (n = 17/35); AMK, 20.0% (n = 7/35); CIP, 62.9% (n = 22/35); and ATM, 38.1% (n = 8/21). Resistance to CIP, CAZ, and FEP was detected in a large proportion of the CRPA isolates (62.9%, 60.0%, and 57.1%), respectively (Table 1).

### 2.3. Genotyping Results Using MLST Analysis

Traditional MLST analysis using the Pasteur nomenclature in the PubMLST database revealed six different known STs and the highest number of unknown STs (27 singletons) among the 35 CRPA sequences analyzed, as shown in Figure 2. The international high-risk clones ST111, ST357, and multiple known STs such as ST1822, ST389, ST3753, and ST175 were identified. Additionally, two identical STs (ST3753) were identified in CRPA isolates originating from the medicine ward of Hospital A. Furthermore, another two identical STs (ST175) were reported from CRPA strains originating from the medicine ward of Hospital C. Hospitals A, B, and C represented the following proportions of unknown STs: n = 6/8, n = 12/16, and n = 9/11, respectively. The reported unknown STs were delineated using Pasteur nomenclature within the Pathogenwatch database. The following results were obtained: 77.1% (n = 27/35) of CRPA isolates were singleton STs with novel alleles, as indicated by an asterisk in Appendix A. In summary, 55.6% (n = 15/27) of the singleton new STs possessed new alleles, while 25.9% (n = 7/27) possessed multiple alleles (7321) at the *trpE* locus. The multi-allelic strains of CRPA are: P08L, P11L, P12L, P15L, and P16L, obtained from Hospital B, and the strains P10C and P12C obtained from Hospital C. Delineation of unknown STs occurs because Pathogenwatch employs a more comprehensive approach to genomic data analysis, allowing it to assign novel STs based on allele matches that may not be present in the PubMLST database. The ability to classify unknown STs as novel STs has significant implications for tracking pathogen evolution and dissemination. Identifying these novel types can help in monitoring emerging strains that may pose public health threats, particularly in the context of AMR. The minimum spanning tree that was constructed using the traditional MLST utilizing seven housekeeping genes revealed the 35 CRPA isolates investigated are largely unique, with the exception of isolates P01T and P05T, and P01C and P05C having identical STs, as shown in Figure 2.

In Figure 2, the inner circles in black represent the first CRPA isolate obtained from each hospital in August 2022, which marked the beginning of the collection from the three hospitals. The first CRPA isolates collected from each hospital were P01T, P01L, and P01C. The circles labeled with more than one strain indicate that these samples exhibited identical STs, such as P01T and P05T, which are represented by ST3753 and P01C, and P05C, which is represented by ST175.

There are CRPA strains that have been observed connected and linked to P16L, represented by novel ST651b, as shown in Figure 2. The strain P15L is represented by another novel STab6a, which is the nearest or closest strain to P16L in terms of allele difference at locus *ppsA*, 14 versus 33, respectively. Both strains are obtained from the intensive care unit five (ICU5) of Hospital B. Additionally, strain P06L, represented by novel ST95c6, obtained from ICU3 of the same hospital, has allele differences in the three loci, namely, *aroE*, *ppsA*, and *trpE*, respectively. The complete allelic profile is shown in Appendix A. In summary, the CRPA strains that have been observed connected and linked to P16L are unique and do not possess clonal structures (Figure 2). Traditional MLST typically analyses a limited number of housekeeping genes, namely, *acsA*, *aroE*, *guaA*, *mutL*, *nuoD*, *ppsA*, and *trpE*, which may not provide sufficient resolution to differentiate closely related strains. Moreover, the high-resolution MLST analysis is crucial in identifying outbreak strains and understanding transmission dynamics.

### 2.4. The Phenotypic and Genotypic Profile of 35 CRPA Isolates

Figure 3 summarizes the phenotypic and genotypic profile of 35 non-duplicate CRPA isolates including the information of the sources of isolates and STs of each strain to understand the epidemiologic landscape of the 35 CRPA strains obtained from the three hospitals. The 35 non-duplicate CRPA isolates obtained from three tertiary hospitals in Metro Manila from August 2022 to January 2023 exhibited high phenotypic and genotypic heterogeneity composed of a large proportion of new strains or novel STs. Moreover, carbapenem resistance profiles are found not associated with specific genetic lineages. The comparative analysis between the genotypic AMR and phenotypic expression will be discussed in detail in subsequent sections.

### 2.5. Presence of Beta-Lactamases

Isolates P05L and P06L, obtained from patients admitted in ICU3, and isolates P14L and P15L, obtained from patients in ICU5 of Hospital B, harbored carbapenemase VIM-2 coproduced with KPC-2 and OXA-74, which was positive for IncP-6 and showed resistance against multiple antibiotics such as imipenem, meropenem, ceftazidime, cefepime, and ciprofloxacin (Figure 3). In addition, isolates P02C and P07C, obtained from patients in Hospital C, harbored beta-lactamases, namely VIM-2 (except for P02C), coproduced with the TEM-1, ESBL CTX-M-15, and OXA-1. OXA-10 also revealed resistance against multiple antibiotics such as imipenem, meropenem, ceftazidime, cefepime, ceftazidime-avibactam, piperacillin-tazobactam, and ciprofloxacin. Additionally, isolate P11C obtained from the medicine ward of Hospital C harbored carbapenemases NDM-7 and TEM-1, coproduced with ESBL CTX-M-15 and OXA-395, which revealed positive for IncX3 and showed resistance against multiple antibiotics, except for ceftazidime-avibactam, as shown in Figure 3. Notably, the majority of isolates did not possess beta-lactamase genes to explain their carbapenem resistance, indicating alternative mechanisms may be at play.

### 2.6. The Presence of Plasmid-Mediated Quinolone Resistance (PMQR) Genes and Mutations in the Quinolone Resistance-Determining Regions (QRDRs) of the gyrA and parE Genes

Of the 35 CRPA isolates, P01C, P05C, P05L, P14L, and P15L carried *gyrA* mutations, while isolates P06L carried both *gyrA* and *QnrVC1* mutations. Two CRPA isolates, namely P02C and P07C, have both (*AAC-(6′)-Ib-cr*) and *gyrA* mutations; isolate P11C have both *gyrA* and *QnrS1* mutations; isolate P12C has *QnrVC1* mutations. Two CRPA isolates, namely P02T and P04T, with both *gyrA* and *parE* mutations, express phenotypic resistance against ciprofloxacin. Moreover, 10 CRPA strains expressed phenotypic resistance to ciprofloxacin, even in the absence of PMQR genes and mutations in the QRDRs (Figure 3). The mutations in the QRDRs include the T83I SNP, which was identified in the *gyrA* gene, while the A473V SNP was identified in the *parE* gene in 31.4% (n = 11/35) and 5.7% (n = 2/35) of the CRPA isolates investigated. The T83I and A473V mutations are known to cause resistance to fluoroquinolones.

### 2.7. Presence of Aminoglycoside Resistance Genes

Isolates P05L, P15L, and P16L showed resistance against amikacin and carried at least three of these aminoglycoside resistance genes: *ANT-(3″)-IIa, ANT(2″)-Ia, APH(6)-Id, APH(3″)-Ib, APH(3′)-IIc, AAC(3)-IIe, AAC(6′)-Ib10, (AAC-(6′)-Ib-cr, AAC(6′)-Ib3, APH(3″)-Ib, APH(3′)-IIc*, and *APH(9)-Ic*. Isolate P11C showed intermediate AST results against amikacin and carried *aadA2*. However, isolates P11L, P12L, P13L, and P16L showed resistance to amikacin, but no resistance genes were identified against amikacin (Figure 3). Further studies on resistance mechanisms are suggested.

### 2.8. Antimicrobial Resistance Genes Flanked by Mobile Genetic Elements (MGEs)

The presence of multiple drug-resistant CRPA strains and multiple antibiotic-resistant genes that are flanked by MGEs poses a potential threat in healthcare settings. In this study, eight AMR genes were flanked by MGEs. These are *bla*_VIM-2_, *bla*_KPC-2_, *bla*_NDM-7_, *bla*_TEM-1_, *bla*_CTX-M-15_, *bla*_OXA-74_, *bla*_OXA-1_, and *bla*_OXA-10_. The *bla*_KPC-2_ in isolates P05L, P06L, P14L, and P15L are flanked by ISKpn6, ISPay42, and ISBma3, while the *bla*_VIM-2_ and *bla*_OXA-74_ are flanked by TnAs3. The *bla*_NDM-7_ in P11C is flanked by IS5D, ISAba125, and ISMpo4. Isolates P02C, P07C, and P11C harbored *bla*_TEM-1_ flanked by ISEc63, Tn2, Tn5393, and IS15DIV, whereas *bla*_CTX-M-15_ is flanked by ISEcp1 and Tn2. Additionally, the *bla*_OXA-10_ in isolates P02C and P07C are flanked by TnAs3 and ISAba43, as shown in Figure 4.

### 2.9. Presence of OprD Mutations

The 28 carbapenemase-negative *P. aeruginosa* isolates were classified based on the three major OprD types. The classifications were made based on the pattern of mutations identified, as seen in Table 2. The OprD type 1 group are CRPA isolates showing “full-length type” OprD. OprD type T1 has been subdivided into 11 subtypes based on the amino acid pattern alterations, including those isolates showing a wild-type PA01 OprD and isolates with several variants because of amino acid substitutions. OprD Type 2 is a group of isolates showing a premature termination, while OprD Type 3 is a group of isolates showing frameshift mutations because of nucleotide insertions or deletions. Type 3 OprD mutations were not identified in this study.

Of the 28 carbapenemase-negative *P. aeruginosa* isolates, P01T and P05T belonged to the OprD Type 2 or OprD deficient type due to a premature stop codon with lacking residues 830–1229, which is known to cause carbapenem resistance. Isolates P07T, P08T, P01C, P03C, P05C, P09C, and P03L belonged to the full-length type OprD Type 1-II with several polymorphisms known to cause carbapenem resistance, whereas P01L has several polymorphisms that were not extensively studied and need further studies for its impact on carbapenem resistance. Additionally, isolates P02L, P08L, P16L, and P03T belonged to a full-length type OprD Type1-Ia, with substitution of one amino acid residue known to cause carbapenem resistance. Interestingly, carbapenemase-negative *P. aeruginosa* isolates P02T, P08C, P10C, P12C, P11L, P12L, P13L, P07L, and P09L belonged to OprD Type 1 and exhibited an intact OprD sequence identical to that of the OprD of wild-type PA01. Furthermore, isolates P04L, P06C, P04T, P06T, and P10L exhibited OprD Type 1-I with a full-length type without substitution of amino acid residues that will cause carbapenem resistance. These results indicates that further studies to identify other resistance mechanisms against carbapenem antibiotics are needed.

## 3. Discussion

This study reported the international high-risk clones ST111, ST357, and multiple STs related to carbapenem-resistant *P. aeruginosa* (CRPA) (ST1822, ST389), including identical STs, namely ST3753, found in Hospital A, and ST175, found in Hospital C. This indicates a significant and widespread issue requiring urgent action. Previous studies reported that in hospital settings, epidemic outbreaks of *P. aeruginosa* MDR high-risk clones are usually associated with ST111 and ST175 [14,23,24]. The international high-risk clone ST357 is persistently identified in the Philippines’ healthcare setting. The Department of Health’s (DOH) Antimicrobial Resistance Surveillance Reference Laboratory (ARSRL) 2022, reported ST357 being the most common [19]. ST357 was also reported in ARSRL, 2023, along with ST485 and ST3986 [20].

The ability of *P. aeruginosa* to produce beta-lactamases, specifically carbapenemases, is a global problem. This article highlights the first report in the Philippines on the identification of *P. aeruginosa* harboring KPC-2, coproduced with VIM-2 and OXA-74. Notably, this is also the first report of KPC in the Philippines identified in *P. aeruginosa*; generally, NDM is the most prevalent MBL identified in the Philippines’ healthcare settings [25]. Similar to published studies, here, we also identified NDM-7, coproduced with CTX-M-15 and TEM-2 from a novel strain ST4b1c obtained from sputum. An article published in 2023, entitled “A study on clinical beta-lactam-resistant gram-negative bacilli in a Philippine tertiary care hospital”, reported *P. aeruginosa* isolates harboring NDM-1, TEM-5, CTX-M-117, IMP-4, OXA-1, and OXA-72, and another isolate harboring NDM-1, IMP-1, OXA-1, and OXA-72 [25]. Additionally, the DOH, ARSRL, 2022, reported IMP-4 from a *P. aeruginosa* isolate obtained from an in-line catheter [19], whereas the DOH, ARSRL, 2023, reported VIM-2 from *P. aeruginosa* obtained from wound discharge [20]. Published reports have also confirmed geographic differences in carbapenemase genotypes; the U.S. epidemic was linked to KPC (class A in the Ambler classification) [12], the European epidemic to VIM (class B) and OXA-48-like (class D), and the Asian epidemic to NDM and IMP (class B) [26].

Carbapenemase and ESBL genes are known to be distributed by MGEs, such as insertion sequences and transposons, accelerating the spread and acquisition of ARGs [27,28]. The presence of multiple drug-resistant CRPA strains and multiple antibiotic-resistant genes that are flanked by MGEs poses a potential threat in healthcare settings. In our study, KPC-2 was flanked by ISPsy42 and ISKpn6; VIM-2 was flanked by TnAs3; and NDM-7 was flanked by IS5D and ISMpo4, whereas other studies reported that KPC-2 genes were flanked by ISKpn6-like and ISKpn8-like MGEs [29]. These modifications can facilitate the movement of resistance genes between different DNA molecules [30].

Published studies showed that ST175 frequently produces VIM-2, while ST111 mostly produces KPC-2 carbapenemase [23]. However, this was not the case in our investigation because ST175 and ST111 did not contain any carbapenemases or ESBLs; instead, ST389 and novel STs, namely ST95c6, STf555, and STab6a, were the strains responsible for the coproduction of VIM-2 and KPC-2. Another investigation underscored the global clinical implications of the CRPA, noting that its genetic and epidemiological attributes are poorly understood [31]. Here, we identified that carbapenem resistance profiles are found to be not associated with specific genetic lineages. Furthermore, the varying resistance profiles of each hospital are due to factors such as local antibiotic use from each hospital, patient demographics, and infection control practices of the three hospitals.

In this study, we determined that seven of the 35 CRPA isolates produced carbapenemases, while 28 are non-carbapenemase-producing strains. Interestingly, the 35 isolates investigated showed phenotypic resistance to at least one antibiotic in three or more antimicrobial classes tested, and 14 of the 28 carbapenemase-negative *P. aeruginosa* were identified as having OprD mutations known to cause carbapenem resistance. Furthermore, we identified 14 carbapenemase-negative *P. aeruginosa* that exhibited an intact OprD. The resistance mechanism of these 14 carbapenemase-negative *P. aeruginosa* against carbapenem could be attributed to outer membrane porin M (OprM) or down-regulation of OprD or other porins, and can also be mediated by the overexpression of drug efflux pumps [4,5,32]. However, efflux pump driven resistance and down regulation of outer membrane porins were not investigated in this study. Additionally, the genetic heterogeneity of the 35 CRPA isolates we investigated contrasts with the clonal dominance of high-risk clones reported in other published studies [23,33,34].

## 4. Materials and Methods

### 4.1. Isolation and Collection of P. aeruginosa

From August 2022 to January 2023, 40 non-duplicate CRPA out of 306 *P. aeruginosa* isolates were obtained from three tertiary hospitals in Metro Manila, Philippines. The three hospitals have been selected because these three hospitals are all private tertiary hospitals located within Metro Manila, showing an increasing incidence of MDROs. The Vitek^®^2 system (bioMérieux, Marcy LE’toile, France) and/or BD Phoenix™ M50 instrument (Becton, Dickinson and Company, Franklin Lakes, NJ, USA) were used for species identification, and the strains that showed imipenem and/or meropenem resistance with a minimum inhibitory concentration (MIC) breakpoint of ≥8.0 mg/L as the result of AST, using the Vitek^®^2 system (bioMérieux, Marcy L E’toile, France) and/or BD Phoenix™ M50 (Becton, Dickinson and Company, USA) instrument, were collected. *P. aeruginosa*, for which the MICs breakpoints were ≥8.0 mg/L for carbapenem (imipenem and/or meropenem), were designated CRPA and were included in the study, provided it was unique or non-duplicated and the source of the CRPA isolates were the following: lower respiratory tract (sputum, endotracheal aspirate (ETA), and bronchoalveolar lavage), wound discharge, blood, and other sterile body fluids. Other sources, such as stool, throat swabs, and nasal swabs, were excluded from the collection. Out of 40 non-duplicate CRPA isolates, 36 were successfully subcultured, and pure colonies were isolated for DNA extraction. Thirty-six CRPA strains were subsequently subjected to WGS; one sample was removed due to low-quality sequence reads, and 35 were included in downstream analysis.

### 4.2. Antimicrobial Susceptibility Test

The AST was performed using the GN N261 CARD of the Vitek^®^2 system (bioMérieux) for the following antibiotics: imipenem (IMP), meropenem (MEM), cefepime (FEP), ceftazidime (CAZ), ciprofloxacin (CIP), piperacillin-tazobactam (TZP), and amikacin (AMK). In addition, the minimum inhibitory concentrations (MICs) of colistin (COL) and ceftazidime-avibactam (CZA) were determined using the Sensititre system™ (Thermo Fisher Scientific™, Waltham, MA, USA) on a EURGNCOL plate via a commercial broth microdilution kit and standard broth microdilution (BMD) according to the CLSI M100 ED32:2022 guidelines [35]. The disc-diffusion method, as outlined in Clinical Laboratory Standard Institute (CLSI) M100-ED32:2022, was utilized for determining antimicrobial resistance against aztreonam (ATM) [35]. The antimicrobial susceptibility of the strains was determined according to the guidelines and breakpoint of the CLSI, M100-ED32:2022, with *P. aeruginosa* ATCC 27853 used as the quality control strain [35].

### 4.3. Whole-Genome Sequencing

The workflow for genomic analysis began with the extraction of genomic DNA (gDNA) from 36 non-duplicate CRPA isolates, which were grown overnight at 37 °C in tryptic soy broth (TSB). The gDNA was extracted using the DNeasy^®^ Ultraclean^®^ Microbial Kit (QiagenTM, Germantown, MD, USA). After extraction, the gDNA was quantified to ensure sufficient yield for downstream applications. The DNA was then fragmented and size-selected to prepare it for library construction. Library preparation was performed using the Nextera XT DNA Library Preparation Kit (Illumina, Inc., San Diego, CA, USA) according to the manufacturer’s protocol. Finally, sequencing was conducted at SA Pathology (Adelaide, Australia) using the Illumina NextSeq 550 platform, employing 150-base paired-end read sequencing. This streamlined workflow highlights the key steps from gDNA extraction to next-generation sequencing (NGS).

### 4.4. Data Processing and Analytics

The raw sequence data were uploaded to an in-house bioinformatics pipeline called EpiTomas v1.0.0 (https://github.com/abulenciamiguel/EpiTomas, accessed on 26 February 2023) [36]. The raw sequences were subjected to read trimming and quality assessment using Fastp (version 0.20.1), ensuring a minimum Phred score of 30 [31]. After preprocessing, the reads were aligned against the reference sequence *P. aeruginosa* PA01 (NCBI nucleotide accession: NC_002516.2) using snippy (version 4.6.0), with a minimum depth of 50 and minimum proportion of 0.9 to be considered as a variant, and the alignment coverage was determined using mosdepth (version 0.3.2) [37]. Additionally, the prediction effect and annotation of the called variants were performed using SnpEff (version 5.1) [38]. Furthermore, plasmid contigs of the 35 CRPA isolates were assembled from the trimmed fastq files using plasmidSPAdes (version 3.15.5) using default parameters [39].

### 4.5. Identification of AMR Genes, Plasmid Replicons, and Flanking MGEs

The consensus sequences from the reference-based alignment and the plasmid contigs were subjected to further analysis, such as the identification and characterization of AMR genes using the Resistance Gene Identifier (RGI, version 6.0.2) with the Comprehensive Antibiotic Resistance Database (CARD, version 3.3.0) as a reference [40]. The 28 carbapenemase-negative *P. aeruginosa* strains that exhibited resistance to imipenem and/or meropenem were subjected to further investigation by identifying OprD mutations that are known to cause carbapenem resistance. Mobile genetic elements (MGEs) in plasmid contigs were further analyzed using the ISfinder web tool (https://isfinder.biotoul.fr, accessed on 30 May 2024) [22]. Furthermore, the plasmid replicons were identified using the PlasmidFinder database (https://cge.food.dtu.dk/services/PlasmidFinder, accessed on 20 May 2024) [41].

### 4.6. Genotyping Using Traditional MLST Analysis

*P. aeruginosa* MLST profiles were predicted using the Pasteur nomenclature, utilizing both the PubMLST [21] and Pathogenwatch databases [42].

### 4.7. Phylogenetic Analysis

The minimum spanning tree was constructed using the GrapeTree MSTreeV2 [43] algorithm to assess *P. aeruginosa* clonality based on allele differences and determine the epidemiological landscape. In addition, IQ-TREE2 [44] was used to construct a maximum likelihood (ML) tree with 1000 bootstrap replicates to assess *P. aeruginosa* genetic diversity based on single nucleotide mutations. Reads from all sequence samples (fastq files) are deposited and available under the BioProject accession PRJNA1113378 and will be made public on 1 March 2025.

## 5. Conclusions

The persistent detection of NDM in the Philippines’ healthcare settings pose a severe global public health threat. Furthermore, this report marks the first identification of a CRPA strain harboring KPC, indicating the inception of potential dissemination of these resistance determinants within *P. aeruginosa* in the Philippines. Continued surveillance and genetic characterization of CRPA isolates are crucial for improving infection control measures aimed at reducing the incidence and spread of CRPA infections by providing current information.

## Figures and Tables

**Figure 1 antibiotics-14-00362-f001:**
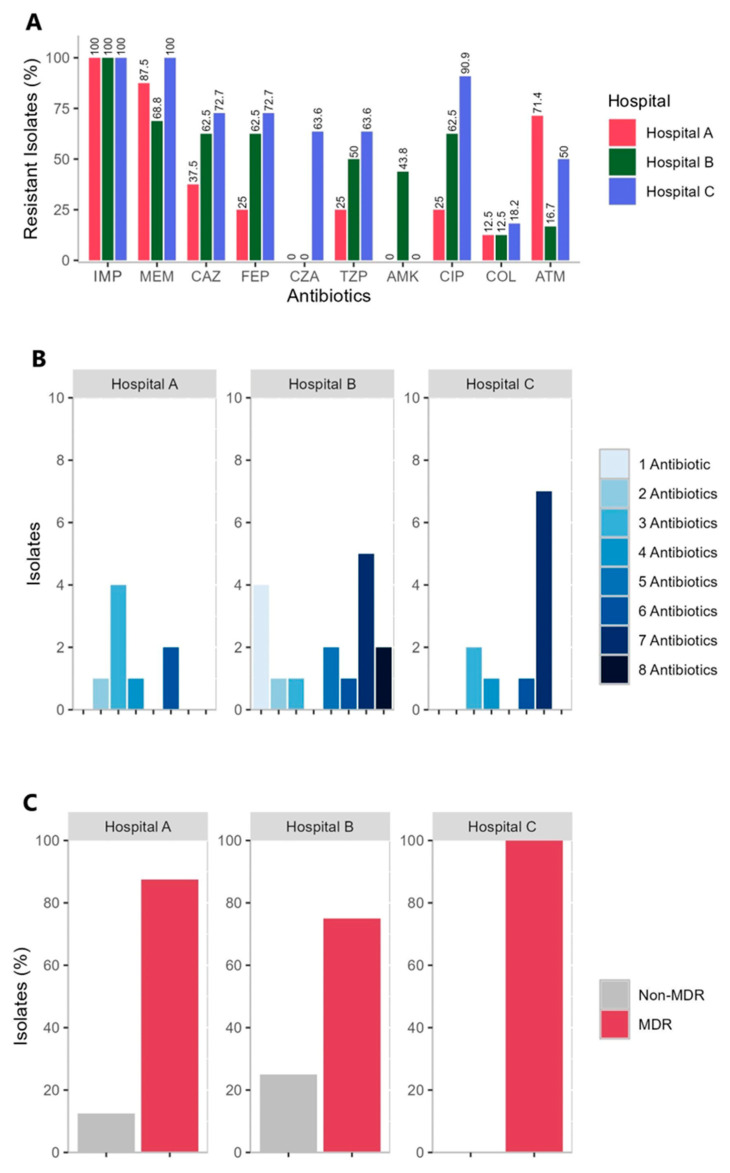
(**A**) Percentage of resistant isolates in the three hospitals. IMP, imipenem; MEM, meropenem; CAZ, ceftazidime; FEP, cefepime; CZA; ceftazidime-avibactam; TZP, piperacillin-tazobactam; AMK, amikacin; CIP, ciprofloxacin; COL, colistin. (**B**) Prevalence of CRPA isolates (n = 35) in each hospital and the number of antibiotics (n = 8) each isolate was resistant to. (**C**) The number of MDR and non-MDR CRPA isolates in each hospital. MDR, multidrug-resistant; Non-MDR, Non-multidrug-resistant.

**Figure 2 antibiotics-14-00362-f002:**
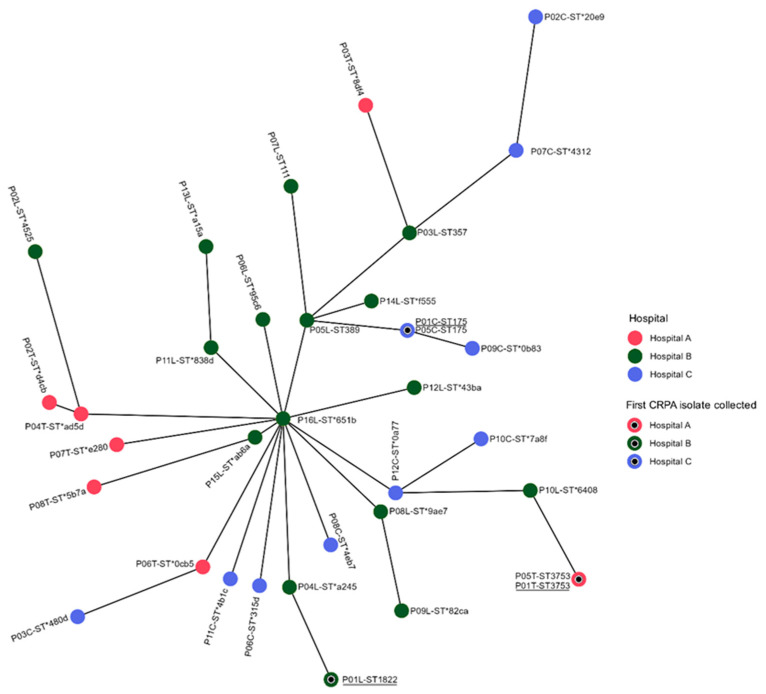
Minimum spanning tree of the 35 non-duplicate CRPA isolates analyzed based on the traditional MLST scheme utilizing the Pasteur nomenclature in the Pathogenwatch and PubMLST databases [21]. The circles in three different colors represent the hospitals. The inner circles in black with underline ST represent the first CRPA isolate obtained from each hospital in August 2022. The circles are named according to the strains and colored according to the hospital’s source. Each circle includes the ST. Asterisk (*) represent novel STs.

**Figure 3 antibiotics-14-00362-f003:**
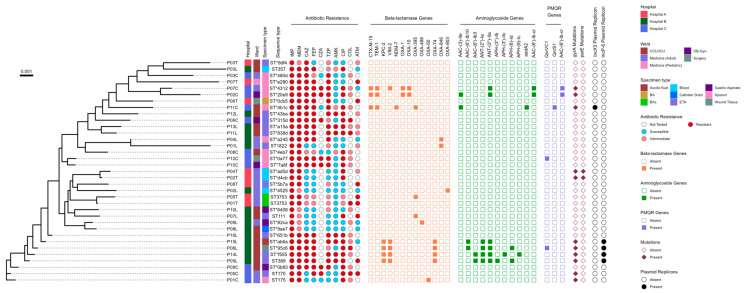
Phylogenetic and SNP analysis of 35 CRPAs obtained from three hospitals (Hospitals A, B, and C) in Metro Manila, Philippines, from August 2022 to January 2023; the corresponding resistance profiles, presence of AMR determinants, and specimen type. SNP analysis was performed using wild-type (WT) PA01 (accession number: NC_002516.2) as the reference genome. The scale bar shows the genetic distinction between each CRPA isolate. The vertical columns indicate from left to right: hospital, patient’s ward/room, specimen type, sequence type (* represent novel STs), resistance phenotype, beta-lactamase resistance genes present (orange filled squares) and absent (empty squares), aminoglycoside resistance genes present (green filled squares) and absent (empty squares), plasmid-mediated quinolone resistance (PMQR) (present = purple filled square; absent = empty square) of acquired resistance genes, and predominant plasmid replicon types (present = black filled circle; absent = empty circle). IMP, imipenem; MEM, meropenem; CAZ, ceftazidime; FEP, cefepime; CZA, ceftazidime-avibactam; TZP, piperacillin-tazobactam; AMK, amikacin; CIP, ciprofloxacin; COL, colistin; ATM, aztreonam; ICU, intensive care unit; CCU, critical care unit; OB-Gyn, obstetrics and gynecology; BAL, bronchoalveolar lavage; BA, bone aspirate; ETA, endotracheal aspirate; and PMQR, plasmid-mediated quinolone resistance.

**Figure 4 antibiotics-14-00362-f004:**
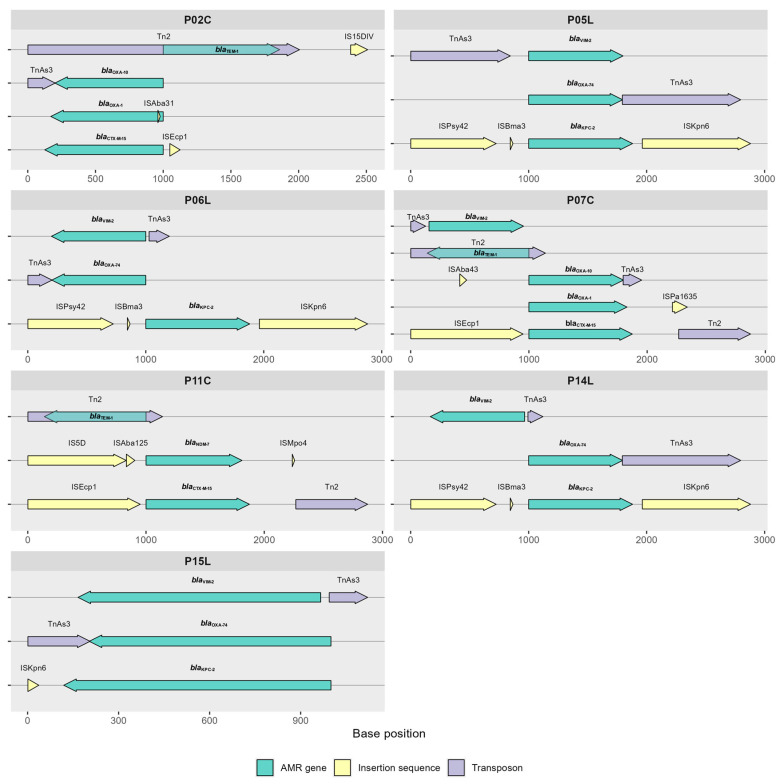
Schematic diagram of eight AMR genes from seven strains of CRPA flanked by MGEs that were analyzed using the ISfinder web tool [22]. AMR genes, insertion sequence, and transposon are represented by three different colors. Numbers are the regions or base positions.

**Table 1 antibiotics-14-00362-t001:** MIC distributions for 35 non-duplicate CRPA isolates obtained from three tertiary hospitals in Metro Manila from August 2022 to January 2023 following the CLS1 2022 guidelines.

Antibiotics	No. of Isolates at an MIC (mg/L) of:												
	0.125	0.250	0.5	1	2	4	8	16	32	64	128	Breakpoint(s) ^b^	R (%)
IMP ^d^,								35 ^+,a^				≤2/≥8	100.0
MEM ^d^						6	3 ^+,a^	1, 25 ^+,a^				≤2/≥8	82.9
CAZ ^d^						11		3, 7 ^+^	1^+,a^	13 ^+,a^		≤8/≥32	60.0
FEP ^d^				5 *	3		3	4, 6 ^+^	1 ^a^	13 ^+,a^		≤8/≥32	57.1
CZA ^e^					1	3	7 ^+^					(≤8/4)/(≥16/4)	63.6
TZP ^d^						4 *	1	4	9	12 ^+^	5 ^+,a^	(≤16/4)/(≥128/4)	48.6
COL ^e^				9 *	6	4 ^a^		1 ^+,a^				≤2/≥4	25.0
CIP ^d^	1 *	4 *	6 *	2	10 ^+,a^	11 ^+,a^				1 ^+,a^		≤0.5/≥2	62.9
AMK ^d^					10 *	3	9 *	3	3	7 ^+,a^		<16/≥64	20.0
	No. of isolates at zone diameter (mm) ^b^ of:												
	12	13	14	15	16	17	18	19	20	21	22	Breakpoint(s) ^c^	
ATM ^f^	1 ^a^			7 ^a^	1	1			4		7	≥22/≤15	38.1

CRPA, carbapenem-resistant *P. aeruginosa*; IMP, imipenem; MEM, meropenem; CAZ, ceftazidime; FEP, cefepime; CZA, ceftazidime-avibactam; TZP, piperacillin-tazobactam; AMK, amikacin; CIP, ciprofloxacin; COL, colistin; and ATM, aztreonam. ^a^ indicates data for resistant strains as per CLSI 2022; * MIC ≤ the indicated value; ^+^ MIC > the indicated value; ^b^ Breakpoint concentrations—expressed as “susceptibility (mg/L)/resistance (mg/L)” and ^c^ Breakpoint concentrations—expressed as “zone of inhibition in mm”—are presented according to CLSI 2022; ^d^ MIC detected using the Vitek^®^2 system (bioMérieux); ^e^ MIC detected using Thermo Fisher Scientific™ Sensititre™; and ^f^ MIC detected using the Disc Diffusion technique.

**Table 2 antibiotics-14-00362-t002:** OprD mutations identified in 28 carbapenemase-negative CRPA isolates.

OprD Type(s)	Resistance Phenotype(s) to IPM	Resistance Phenotype(s) to MEM	Isolates	OprD Type Classification	Mutations	Known to Cause Carbapenem Resistance	OprD Structure(s) Affected
**T1**	Resistant	Resistant	P02T, P08C, P10C, P12C, P11L, P12L, P13L	Full-length type, Wild-type	None	None	None
**T1**	Resistant	Intermediate	P07L, P09L	Full-length type, Wild-type	None	None	None
**T1-I**	Resistant	Resistant	P04L, P06C, P04T, P06T	Full-length type, without substitution of amino acid residue	None	None	None
**T1-I**	Resistant	Intermediate	P10L	Full-length type, without substitution of amino acid residue	None	None	None
**T1-Ia**	Resistant	Intermediate	P02L	Full-length type, substitution of 1 amino acid residue	Phe205Ile	Yes	Alters hydrophobicity and stability of the OprD channel
**T1-Ia**	Resistant	Intermediate	P03T	Full-length type, substitution of 1 amino acid residue	Val379Leu	Yes	Affects pore size and shape, reducing drug entry
**T1-Ia**	Resistant	Resistant	P16L	Full-length type, substitution of 1 amino acid residue	Ile628,629Ala	Yes	Impacts structural integrity and function of the OprD channel
**T1-Ia**	Resistant	Intermediate	P08L	Full-length type, substitution of 1 amino acid residue	Ile628,629Ala	Yes	Impacts structural integrity and function of the OprD channel
**T1-II**	Resistant	Resistant	P08T	Full-length type, with several polymorphisms	Lys344Thr, Phe508Leu	Unknown	None
**T1-IIa**	Resistant	Resistant	P05C	Full-length type, with several polymorphisms	Glu604Gln, Ile628,629Ala	Yes Yes	Affects L5, Impacts structural integrity and function of the OprD channel.
**T1-IIb**	Resistant	Resistant	P03C	Full-length type, with several polymorphisms	Glu688Lys, Ser719Thr	Yes Unknown	Affects L6 and electrostatic interactions, altering pore dynamics
**T1-III**	Resistant	Resistant	P07T, P09C, P01C	Full-length type, with several polymorphisms	Glu604Gln Ile628,629Ala Glu688Lys Ser719Thr	Yes Yes Yes Unknown	Affects L5, L6, structural integrity and function of the OprD channel, and electrostatic interactions, altering pore dynamics
**T1-IIIa**	Resistant	Resistant	P01L	Full-length type, with several polymorphisms	Thr308Ser Arg928,929Glu Ala944,945Gly Gly1,274Ala	Unknown Unknown Unknown Unknown	None
**T1-IV**	Resistant	Resistant	P03L	Full-length type, with several polymorphisms	Ser169,170,171Glu Ser175,177Arg Val379Leu Glu688Lys Ser719Thr Gln1,270Glu	Yes Yes Yes Yes Unknown Unknown	Affects L1, L3, L6, L8, altering pore characteristics
**T2**	Resistant	Resistant	P01T, P05T	OprD deficient Type—Premature stop codon	Trp830 * (W830X)	Yes	Loss of porin function

OprD, outer membrane porin D; L, designates any of the OprD loops possibly affected by a determined amino acid substitution. Phe, Phenylalanine; Ile, Isoleucine; Val, Valine; Leu, Leucine; Glu, Glutamate; Ala, Alanine; Lys, Lysine; Gln, Glutamine; Ser, Serine; Thr, Threonine; Arg, Arginine; Trp, Tryptophan; *, an undetermined amino acid.

## Data Availability

Reads from all sequence samples (fastq files) are available under the BioProject accession PRJNA1113378 and will be made public on 1 March 2025.

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
