# Peer review of "Identification of a Potential High-Risk Clone and Novel Sequence Type of Carbapenem-Resistant Pseudomonas aeruginosa in Metro Manila, Philippines"

_antibiotics, 2025, doi:10.3390/antibiotics14040362_

Round 1
Reviewer 1 Report
Comments and Suggestions for Authors
The manuscript is a well-structured and significant contribution to the field of antimicrobial resistance, particularly in understanding carbapenem-resistant Pseudomonas aeruginosa (CRPA) in the Philippines. It provides valuable insights into the genetic diversity, resistance mechanisms, and epidemiology of CRPA strains, including novel findings like the detection of KPC-2, which is being reported for the first time in the region.
The methodology is robust, with detailed descriptions of genomic analysis, antimicrobial susceptibility testing, and sequence typing. The results are comprehensive, highlighting critical public health concerns and emphasizing the need for continued surveillance and infection control.
However, the manuscript would benefit from improved clarity and conciseness, particularly in the Abstract, Introduction, and Discussion sections. Some sentences are overly complex, and redundancy should be minimized. Visual aids and figure legends could be more informative to enhance accessibility for readers. Addressing these minor issues will further strengthen the manuscript's impact and readability.
The overuse of "interestingly" dilutes its emphasis. Alternatively, use “Notably” or other words if emphasis is needed.
Line 29 – Consider to change: Six sequence types (STs), including the international high-risk clones ST111 and ST357, were identified.
Line 35 – Consider change to: The relentless identification of NDM in the Philippine healthcare setting poses a significant global public health risk.
Lines 61-64 - Sentence is overly long and complex. Consider change to: The acquisition of carbapenemases is critical because these enzymes can hydrolyze most beta-lactam antibiotics. Carbapenemase genes are often carried on mobile genetic elements, such as plasmids, transposons, and insertion sequences, which facilitate their transfer to other bacterial species.
Line 156- 159 Consider to change: The following results were obtained: 77.1% (n = 27/35) of CRPA isolates were singleton STs with novel alleles, as indicated by an asterisk in Supplementary Table S3.
Line 236 – gyrA mutations
Overall, this is a strong and relevant study with valuable implications for addressing the growing threat of antimicrobial resistance in a global context.
Author Response
Thank you for your comments.
Please see the attached file provided for a point - by-point response to your comments.

Reviewer 2 Report
Comments and Suggestions for Authors
Firstly, I want to congratulate the authors for the hard work that they have done. Here I have made some comments:
Line 1-5: Title is quite big, make it small and precise like “Identification of a potential high-risk clone and novel sequence type of carbapenem-resistant Pseudomonas aeruginosa in Metro Manila, Philippines
Line 54-65: Write a brief about KPC-2, VIM-2 and OXA-74 and why these are important
Line 70: Use acronym of antimicrobial resistance (AMR)
Line 71: Use AMR instead of antimicrobial resistance
Line 306-308: The impact of antimicrobial stewardship programs is mentioned but lacks supporting data. Consider referencing or removing this assertion if data is unavailable
Line 321: Use acronym, Department of Health’s (DOH) Antimicrobial Resistance Surveillance Reference Laboratory, 2022 (ARSRL), uses several times
Line 362: How many samples did you collect? How did you collect the samples?
Line 363: How did you select these three hospitals?
Line 380: Why did you select these antibiotics?
Comments on the Quality of English LanguageEnglish could be improved to more clearly express the research.
Author Response
Thank you for your comments.
I have attached my point-by-point response.

Reviewer 3 Report
Comments and Suggestions for Authors
Some minor comments are provided in the attached file.

Author Response
Thank you for your comments and suggestions.
Please see the attachment that provide a point - by - point response to your comments.
